# Prevalence of Major Bleeding in Elderly Patients on Oral Anticoagulants for Non-Valvular Atrial Fibrillation: A Single-Center 12-Year Retrospective Review

**DOI:** 10.3390/geriatrics10060165

**Published:** 2025-12-15

**Authors:** How Foong Kwan, Hazlina Mahadzir, Nor Rafeah Tumian, Azimatun Noor Aizuddin, Shue Hong Kong

**Affiliations:** 1Department of Medicine, Faculty of Medicine, University Kebangsaan Malaysia, Kuala Lumpur 43600, Malaysia; howfoong21@gmail.com; 2Department of Geriatrics, Faculty of Medicine, University Kebangsaan Malaysia, Kuala Lumpur 43600, Malaysia; 3Department of Haematology, Faculty of Medicine, University Kebangsaan Malaysia, Kuala Lumpur 43600, Malaysia; rafeah@hctm.ukm.edu.my; 4Department of Secretariat of Postgraduate Studies, University Kebangsaan Malaysia, Kuala Lumpur 43600, Malaysia; azimatunnoor@ppukm.ukm.edu.my; 5Department of Pharmacy, University Kebangsaan Malaysia, Kuala Lumpur 43600, Malaysia; shkong@ppukm.ukm.edu.my

**Keywords:** non-valvular atrial fibrillation, direct oral anticoagulants, warfarin

## Abstract

**Background/Objectives:** Non-valvular atrial fibrillation (NVAF) is a common arrhythmia in the elderly and carries a high risk of cardioembolic stroke. Oral anticoagulation is central to prevention, with direct oral anticoagulants (DOACs) increasingly replacing warfarin due to better safety and convenience. However, major bleeding remains a key concern, particularly in older patients. This study aimed to determine the prevalence of major bleeding among elderly patients (≥65 years) with NVAF treated with oral anticoagulants. **Methods:** A retrospective cohort study was conducted on 886 elderly NVAF patients managed at a tertiary hospital between January 2012 and December 2023. Data on demographics, anticoagulant type, comorbidities, and bleeding events were collected. Associations between categorical variables were tested using Chi-square or Fisher’s exact tests, while logistic regression identified predictors of major bleeding. **Results:** The mean age was 78.4 ± 7.2 years, with equal gender distribution. Most patients (87.1%) received DOACs, while 12.9% were prescribed warfarin. A total of 63 patients (7.1%) experienced major bleeding, including 51 (6.6%) in the DOAC group and 12 (10.5%) in the warfarin group. Intracranial and intra-/retroperitoneal hemorrhages were most common. Logistic regression showed older age, prior bleeding, a higher HASBLED score, and antiplatelet use as significant predictors. Among patients with a recorded weight (*n* = 70), dosing adherence was better for apixaban and edoxaban compared to dabigatran and rivaroxaban. **Conclusions:** DOACs were associated with fewer major bleeding events than warfarin. Bleeding risk was strongly linked to age, prior bleeding, HASBLED score, and concomitant antiplatelet therapy, highlighting the importance of appropriate DOAC dosing for safety.

## 1. Introduction

Atrial fibrillation (AF) is the most common sustained cardiac arrhythmia, characterized by disorganized atrial electrical activity and ineffective atrial contraction [1]. Hallmark electrocardiographic features include irregular R-R intervals, absence of discrete P waves, and fibrillatory waves [2]. In the absence of a conventional electrocardiogram (ECG), AF can also be detected using implanted devices such as pacemakers, defibrillators, or loop recorders [3].

Globally, the incidence and prevalence of AF have been rising. Data from the Framingham Heart Study indicate a threefold increase over the past 50 years [4]. The Global Burden of Disease project estimated that 46.3 million people were affected worldwide in 2016, and projections suggest that at least 72 million people will be affected in Asian countries by 2050 [5]. In Malaysia, prevalence is lower at 0.54%, versus the global average of 1% [6]. This trend is notable in the elderly, as the proportion of adults ≥ 65 years in Malaysia is projected to rise from 12% in 2010 to 22% by 2040 [7].

Stroke prevention and systemic thromboembolism remain the cornerstone of AF management. According to the Malaysian Stroke Registry (2009–2016), 3% of stroke patients had AF [8]. The CHA_2_DS_2_-VASc score predicts one-year thromboembolic risk in NVAF, guiding anticoagulation; therapy is considered for scores of 1 and recommended for scores ≥ 2 [9,10,11]. Both warfarin and DOACs—including dabigatran, rivaroxaban, apixaban, and edoxaban—are used for stroke prevention. Despite benefits, both warfarin and DOACs carry bleeding risk. The HASBLED score estimates one-year major bleeding risk [12,13]. Scores 0–1 indicate low risk, a score of 2 indicates moderate, scores 3–5 indicate high risk, and scores > 5 indicate extremely high risk, guiding decisions on anticoagulation or alternatives.

In Japan, the All Nippon AF in the Elderly (ANAFIE) registry [14], a multicenter prospective observational study conducted from October 2016 to January 2020 with a two-year follow-up, reported that among patients aged ≥85 years, the incidence of all bleeding events was lower in the DOAC group (4.28 per 100 person-years) compared to the warfarin group (5.12 per 100 person-years). Similarly, the RE-LY trial (2009) found that the annual rate of major bleeding was 3.36% in the warfarin group versus 2.71% in the dabigatran 110 mg group and 3.11% in the dabigatran 150 mg group [15].

A retrospective study by Villines et al. [16] compared adults with non-valvular atrial fibrillation newly started on standard-dose DOACs. Between July 2011 and June 2016, dabigatran (150 mg twice daily) was compared with rivaroxaban (20 mg once daily), while from January 2013 to June 2016, dabigatran (150 mg twice daily) was compared with apixaban (5 mg twice daily). Patients on dabigatran had a significantly lower risk of major bleeding compared to rivaroxaban (2.08% vs. 2.53%) with similar stroke risk (0.60% vs. 0.78%). No significant difference in major bleeding risk was observed between dabigatran and apixaban (1.60% vs. 1.21%).

Elderly patients pose unique challenges due to their frailty, comorbidities, and care complexity, making anticoagulation a clinical dilemma. Balancing bleeding risk against thromboembolism is critical to optimize outcomes and quality of life. Therefore, this study aims to evaluate the prevalence of major bleeding in elderly patients aged ≥65 years with NVAF receiving oral anticoagulants, including DOACs and warfarin.

## 2. Materials and Methods

### 2.1. Study Design, Population and Methodology

This was a single-center retrospective study among elderly patients with NVAF from Hospital Canselor Tuanku Muhriz (HCTM) from January 2012 to December 2023. Patients administered anti-coagulants for NVAF throughout this period were recruited in this study via a convenient sampling method, which included eligible patients during the study period until a pre-specified precision target was reached. Eligible patients were identified using diagnosis codes and pharmacy dispensing records, and each case was manually reviewed to confirm inclusion and exclusion criteria. Major bleeding events were verified through detailed chart review, including documentation in clinical notes, laboratory evidence (e.g., hemoglobin drop ≥ 2 g/dL or transfusion requirement), and imaging or endoscopic findings consistent with the International Society on Thrombosis and Haemostasis (ISTH) definition of major bleeding. Hence, this was a retrospective cross-sectional review of medical records rather than a longitudinal follow-up study.

The inclusion criteria were elderly patients older than 65 years with underlying NVAF proven by either electrocardiogram/echocardiogram or Holter test, on oral anticoagulants, either DOAC or warfarin. The exclusion criteria were valvular AF and patients taking oral anti-coagulants for indications other than non-valvular atrial fibrillation, such as deep vein thrombosis and pulmonary embolism.

### 2.2. Study Variables

In this study, bleeding events were classified as either major or minor according to the criteria established by the International Society on Thrombosis and Haemostasis. Bleeding episodes unrelated to DOAC or warfarin therapy, such as those provoked by trauma, were excluded. Major bleeding was defined as fatal bleeding, symptomatic bleeding occurring in a critical area or organ (including intracranial, intraspinal, intraocular, retroperitoneal, intraarticular, pericardial, or intramuscular sites), or bleeding that resulted in a hemoglobin decrease of ≥2 g/dL or required transfusion of at least two units of whole blood or red cells.

The study collected multiple variables, including demographic parameters (age, gender, ethnicity), anthropometric data (weight), HASBLED score, comorbidities (hypertension, chronic kidney disease, liver disease, stroke history, or prior major bleeding), concomitant antiplatelet use, type and dose of oral anticoagulants, and reported bleeding events across the specified critical sites. Age in the elderly population was further stratified into young–old (65–74 years), middle–old (75–84 years), and old–old (≥85 years).

Dose appropriateness for each DOAC was assessed according to guideline-recommended criteria. For apixaban, a dose reduction was recommended if at least two of the following were present: age > 80 years, serum creatinine > 1.5 mg/dL, or body weight < 60 kg. For rivaroxaban, a reduced dose of 15 mg once daily was recommended in patients with creatinine clearance (CrCl) < 50 mL/min, and its use was not advised when CrCl was <15 mL/min. For dabigatran, the recommended dose was reduced to 110 mg twice daily in patients aged > 80 years and to 75 mg twice daily for those with CrCl 15–30 mL/min, while use was contraindicated if CrCl < 15 mL/min. For edoxaban, a reduced dose of 30 mg once daily was recommended for patients with CrCl 15–50 mL/min or body weight < 60 kg.

### 2.3. Sample Size

Rather than a priori power calculation, we set a precision target for key proportions (e.g., major bleeding). Assuming event rates around 10–20%, a 95% CI half-width of ~2–3% requires approximately *n* ≈ 550–700 (Wald approximation: *n* ≈ z^2^*p*(1 − *p*)/d^2^ with z = 1.96). We therefore aimed for ≥800 records and ultimately included *n* = 886, which yields ~±2.0% precision if the true proportion is 10% and ~±2.6% if 20%. All estimates are presented with 95% CI.

### 2.4. Ethical Considerations 

The study was conducted in accordance with the Declaration of Helsinki and approved by the Institutional Review Board of Hospital Canselor Tuanku Muhriz (project code HTM-2023-022 and date of approval 13 September 2023).

### 2.5. Data Analysis

Statistical Package for the Social Sciences (SPSS) statistical software for Windows version 29.0 was used for the statistical analysis. Normally distributed numerical variables were reported as mean ± SD, while non-normally distributed variables were presented as median with interquartile range (IQR, 25th–75th percentile). Categorical variables were summarized as frequencies and percentages.

The comparison of major bleeding events between warfarin and direct oral anticoagulants was assessed using Pearson’s Chi-square test or Fisher’s exact test, as appropriate. These tests were also applied to examine the association between HASBLED score and the safety of oral anticoagulants. All *p*-values < 0.05 were considered statistically significant.

Simple and multiple logistic regression analyses were performed to identify factors associated with major bleeding events, including age, gender, concomitant antiplatelet usage, stage of chronic kidney disease, hypertension, stroke history, history of major bleeding and liver disorder. Crude odds ratios (OR) were first calculated to assess the unadjusted association between each predictor and the outcome. Variables with *p* < 0.25 were then included in the multivariable logistic regression to obtain adjusted odds ratios (aOR).

## 3. Results

### 3.1. Study Population

A total of 886 patients were included in the analysis, with their baseline characteristics summarized in Table 1. The mean age was 78.38 ± 7.15 years, with most patients classified as middle–old, followed by young–old and old–old groups. Gender distribution was nearly equal between males and females. Ethnically, Chinese patients were the largest group, followed by Malay, Indian, Punjabi, and other ethnicities.

Body weight was documented for only 70 patients, the majority of whom weighed over 60 kg, with a mean of 66.65 ± 15.90 kg. According to the HAS-BLED score, most patients were in the low- to moderate-risk category (score < 3). Hypertension was the most common comorbidity, followed by a history of stroke, advanced chronic kidney disease, liver disease, and prior major bleeding events. Concomitant use of antiplatelets with oral anticoagulants was observed in only 4 patients.

Most elderly patients with NVAF were prescribed DOACs rather than warfarin. Among the DOACs, rivaroxaban was the most frequently used (31.7%), followed by apixaban, dabigatran, and edoxaban.

### 3.2. Major Bleeding Event

Table 2 presents the occurrence of major bleeding events among elderly patients with NVAF treated with either DOACs or warfarin. A total of 51 patients in the DOAC group and 12 patients in the warfarin group experienced major bleeding. Although the absolute number was higher in the DOAC group, the percentage of major bleeding was lower in the DOAC group (6.6%) compared to the warfarin group (10.5%) due to the larger sample size. The Chi-square test yielded a *p*-value of 0.128, which is greater than 0.05, and the test statistic was below the critical value of 3.84 (df = 1). Therefore, the difference was not statistically significant.

Table 3 summarizes major bleeding events across individual DOACs. The highest number of events occurred in patients on dabigatran and rivaroxaban, followed by apixaban and edoxaban. However, owing to the smaller sample size, edoxaban showed the highest proportion of major bleeding, followed by dabigatran, apixaban, and rivaroxaban. These differences were also not statistically significant, as indicated by the respective test statistics and *p*-values.

Table 4 summarizes the distribution of major bleeding events across different types of DOACs and warfarin. The most frequently observed bleeding types were intracranial, intra-/retroperitoneal, and other bleeding events. A total of 39 intra-/retroperitoneal bleeds were reported, occurring most commonly in patients on apixaban, followed by dabigatran and warfarin, and less frequently with rivaroxaban and edoxaban. For intracranial bleeding, 19 events were observed, predominantly in patients receiving dabigatran, followed by apixaban and rivaroxaban, with fewer cases in edoxaban and warfarin users. Only five other bleeding events occurred, and mainly among rivaroxaban and warfarin patients. None of these differences reached statistical significance.

### 3.3. Predictors of Major Bleeding

Table 5 summarizes the association between various factors and the occurrence of major bleeding. In the unadjusted model, each one-year increase in age was associated with 6% higher odds of major bleeding (*p* < 0.001). Compared with patients aged 65–74 years, those aged 75–84 years had 2.67 times higher odds (*p* = 0.010), while those aged ≥ 85 years had 3.44 times higher odds (*p* = 0.003).

By ethnicity, Punjabi patients had 15.09 times higher odds of major bleeding compared with Malay patients (*p* = 0.008). Patients receiving concomitant antiplatelet therapy had 4.41 times higher odds than those without antiplatelet use, though this was not statistically significant (*p* = 0.202). A history of major bleeding strongly predicted recurrence, with affected patients having 41.10 times higher odds compared with those without such a history (*p* = 0.001). Conversely, patients with a history of cerebrovascular disease had lower odds (OR = 0.55) of major bleeding (*p* = 0.169).

In the multivariable logistic regression model, each additional year of age was associated with a 7% increase in the odds of major bleeding after adjusting for history of prior bleeding (*p* < 0.001). Patients with a history of major bleeding remained at substantially elevated risk, with 55.89 times higher odds compared with those without such a history, after adjustment for age (*p* = 0.001).

### 3.4. Predictor of HASBLED in Major Bleeding

Table 6 presents the relationship between HAS-BLED score and the prevalence of major bleeding among elderly patients receiving oral anticoagulants (both DOAC and warfarin). Of the 886 patients, 29 with a HAS-BLED score < 3 experienced major bleeding, compared with 34 patients with a score ≥ 3. The association was statistically significant (*p* < 0.001).

### 3.5. Dose Appropriateness of DOAC

Table 7 summarizes the dose appropriateness of DOACs in elderly patients with NVAF. Among the cohort, body weight was documented for only 70 patients, while data were missing for 816 patients. This parameter is essential for calculating creatinine clearance (CrCl), a key component in determining DOAC dose appropriateness. Within this subgroup, 88.5% of patients on apixaban were prescribed an appropriate dose, followed by 80% on edoxaban, 64.3% on rivaroxaban, and 54.5% on dabigatran.

## 4. Discussion

This study sets out to investigate the prevalence of major bleeding among elderly patients with NVAF receiving oral anticoagulants, including both warfarin and DOACs. The study population largely consisted of individuals in the “middle–old” category (75–84 years), with a mean age of 78.38 ± 7.15 years and an equal distribution of male and female patients. Notably, the vast majority (*n* = 772, 87.1%) were treated with DOACs, reflecting the shift in contemporary clinical practice. This finding aligns with global prescribing patterns, where DOACs have gradually replaced warfarin as the preferred anticoagulant due to their more predictable pharmacokinetics, fewer food and drug interactions, absence of routine INR monitoring, and improved patient adherence [17].

In this study, the prevalence of major bleeding was 63 events among 886 patients (7.11%). Of these, 51 (6.6%) occurred in patients on DOACs, while 12 (10.5%) occurred in those on warfarin. Although the absolute number of bleeding cases was higher in the DOAC group, this reflects the much larger sample size in that group. Importantly, when expressed as proportions, patients on DOACs experienced fewer bleeding events compared with warfarin users. This observation is consistent with findings from large-scale meta-analyses and multiple international cohort studies [18,19,20], which have consistently shown a more favorable safety profile, such as predictable pharmacokinetics, fewer drug–drug and food–drug interactions for DOACs relative to warfarin, particularly in elderly populations.

This study presented crude proportions of major bleeding events rather than incidence rates adjusted for time at risk, as detailed treatment duration and follow-up time were not consistently available in the medical records. Therefore, patient-year exposure and incidence rate ratios (IRR) could not be calculated. Future studies with complete follow-up data should incorporate person-time analyses to allow more accurate comparison of bleeding risk between DOAC and warfarin users.

Edoxaban, the most recently approved DOAC (FDA approval in January 2015 based on the ENGAGE AF-TIMI 48 trial) [21,22], was the least prescribed agent in our center; only 31 patients received edoxaban, of whom four (12.9%) experienced major bleeding. Due to this limited sample size, the results for edoxaban may not be generalizable, and larger cohorts would be necessary to clarify its bleeding risk profile in our population. In contrast, dabigatran, apixaban, and rivaroxaban had sufficient representation, yet none demonstrated statistically significant differences in major bleeding outcomes. Of note, rivaroxaban appeared to have a lower prevalence of intra-/retroperitoneal and intracranial bleeding compared with dabigatran and apixaban, echoing findings from nationwide observational data, although again, these results were not statistically significant [23].

The most common bleeding events observed were intra-/retroperitoneal, intracranial bleeds and other bleeding. Rare bleeding events such as intraspinal, intraocular, intra-articular, pericardial, and intramuscular bleeding were not documented in our study and may have been either absent or under-reported. These findings underscore the importance of surveillance for both clinically overt and less common bleeding complications when managing anticoagulated elderly patients.

The analysis also revealed a clear relationship between age and the risk of major bleeding. Each additional year of age was associated with a 6% increase in bleeding risk, independent of other factors. This age-dependent rise mirrors findings from pivotal DOAC trials in NVAF, which consistently demonstrate that patients aged ≥80 years are at substantially higher risk of major bleeding compared to those <80 years, likely due to increased frailty, comorbidities, and care complexity. This highlights the need for heightened vigilance and tailored strategies when prescribing anticoagulation to the elderly population [24].

Another important finding from this study was the strong predictive value of a prior history of major bleeding, which markedly increased the likelihood of recurrence due to fragile or diseased blood vessels and age-related vascular changes. This aligns with cohort studies [25,26] demonstrating that patients with a history of spontaneous bleeding, particularly intracranial bleeding, face a persistently elevated re-bleeding risk. Such patients warrant careful individualized risk stratification, and tools like the HASBLED score can assist in clinical decision-making by incorporating prior bleeding, age, renal function, and comorbidities.

Indeed, the higher HASBLED score in our cohort was significantly correlated with bleeding risk. Higher HASBLED score indicates accumulation of more clinical factors that impair vascular integrity, alter hemostasis, or increase drug exposure—all of which synergistically raise the risk of major bleeding. These findings reaffirm the utility of HASBLED as a practical tool in predicting bleeding complications in anticoagulated elderly populations [26,27]. For patients deemed very high risk, particularly those with recurrent bleeding or contraindications to long-term anticoagulation, alternative stroke prevention strategies such as left atrial appendage occlusion devices may be considered.

A prior history of stroke is associated with an increased risk of major bleeding, consistent with the key findings reported in numerous studies [28,29]. A prior history of stroke often indicates a fragile cerebral vasculature. After stroke, chronic inflammation, endothelial dysfunction, and microvascular changes make small vessels more susceptible to rupture under anticoagulant effects.

Beyond age, concurrent use of antiplatelet agents emerged as another factor associated with increased bleeding risk, as the synergistic effect affects both primary hemostasis (platelet plug) and secondary hemostasis (fibrin formation), consistent with pharmacodynamic interactions documented in the literature. Studies have shown that the combination of DOACs and antiplatelets roughly doubles the risk of bleeding compared with DOAC monotherapy [30]. In our study, concurrent antiplatelet use was statistically associated with major bleeding, although only four patients were affected. Current clinical guidance emphasizes minimizing the duration of combined therapy, employing proton pump inhibitors for gastrointestinal protection where appropriate, and reassessing the ongoing need for dual therapy after bleeding events to strike the optimal balance between thrombotic protection and bleeding risk.

Dose appropriateness of DOACs was another dimension explored in this study. Of the 70 patients with recorded weight, higher rates of correct dosing were observed for apixaban (88.5%) and edoxaban (80%) compared with rivaroxaban (64.3%) and dabigatran (54.5%). These differences may reflect both prescriber familiarity and the complexity of dosing criteria across agents. Previous studies have documented frequent deviations from recommended DOAC dosing, with both under-dosing and over-dosing linked to adverse outcomes [31,32,33,34,35]. Possible contributors include prescriber uncertainty about dose-adjustment criteria, time constraints in busy clinics, and physician perception of patient-specific bleeding or thrombotic risk. These observations underscore the importance of ongoing education for prescribers and periodic audit of prescribing practices to ensure optimal anticoagulant use.

## 5. Conclusions

In conclusion, DOACs were associated with a lower observed prevalence of major bleeding compared to warfarin, but the difference was not statistically significant, possibly reflecting the limited sample size and event rate. Age, prior bleeding history, concomitant antiplatelet use, and higher HAS-BLED scores were key factors associated with bleeding risk. These findings reinforce the importance of individualized risk stratification and careful dose selection, particularly considering renal function and potential drug interactions. Pharmacist involvement and regular reviews of medication are recommended to optimize anticoagulant safety in elderly patients. While this study is limited by its single-center design, modest event numbers, and missing weight data, it provides valuable insights into current prescribing practices and highlights the need for larger multicenter studies to validate these observations.

## Figures and Tables

**Table 1 geriatrics-10-00165-t001:** Baseline characteristics of patients with NVAF.

Variables				*n* (%) Total = 886	Mean ± SD/Median (IQR)
Age					78.38 ± 7.15
	65–74			278 (31.4)	
	75–84			414 (46.7)	
	≥85			194 (21.9)	
Gender	Male			424 (47.9)	
	Female			462 (52.1)	
Ethnic group	Malay			354 (40.0)	
	Chinese			496 (56.0)	
	Indian			31 (3.5)	
	Punjabi			4 (0.5)	
	Others			1 (0.1)	
Weight					66.65 ± 15.90
	<60 kg			23 (2.6)	
	≥60 kg			47 (5.3)	
	Missing data			816 (92.1)	
HASBLED	<3			641 (72.3)	
	≥3			245 (27.7)	
Comorbidities	Hypertension			758 (85.6)	
	CKD staging		1–3	833 (94.0)	
			4–5	53 (6.0)	
	Liver disorder			25 (2.8)	
	Stroke history			139 (15.7)	
	History of major bleeding			4 (0.5)	
Medications	Concomitant antiplatelet usage			4 (0.5)	
	Warfarin			114 (12.9)	
	DOAC			772 (87.1)	
		Apixaban		237 (26.7)	
		Dabigatran		223 (25.2)	
		Edoxaban		31 (3.5)	
		Rivaroxaban		281 (31.7)	

**Table 2 geriatrics-10-00165-t002:** Major bleeding in the Direct Oral Anticoagulants (DOACs) and warfarin groups.

	Major Bleeding, *n* (%) Total = 886		Statistical Test	*p* Value
	Yes	No		
DOAC	51 (6.6)	721 (93.4)	2.31 (1) ^a^	0.128
Warfarin	12 (10.5)	102 (89.5)		

^a^ Pearson Chi Square.

**Table 3 geriatrics-10-00165-t003:** Major bleeding in each type of Direct Oral Anticoagulants (DOACs).

	Major Bleeding, *n* (%) Total = 772		Statistical Test	*p* Value
	Yes	No		
Apixaban	15 (6.3)	222 (93.7)	0.30 (1) ^a^	0.584
Dabigatran	16 (7.2)	207 (92.8)	0.002 (1) ^a^	0.966
Edoxaban	4 (12.9)	27 (87.1)	-	0.270 ^b^
Rivaroxaban	16 (5.7)	265 (94.3)	1.25 (1) ^a^	0.263

^a^ Pearson Chi Square; ^b^ Fisher’s Exact test; - Statistical test not needed in Fisher’s exact test.

**Table 4 geriatrics-10-00165-t004:** Types of major bleeding in the Direct Oral Anticoagulants (DOACs) and warfarin groups.

	Oral Anticoagulant, *n* (%)					Statistical Test	*p* Value
	Apixaban	Dabigatran	Edoxaban	Rivaroxaban	Warfarin		
Intra-/Retro-peritoneal	11 (28.2)	9 (23.1)	2 (5.1)	8 (20.5)	9 (23.1)	5.33 (4) ^b^	0.255
Intracranial	4 (21.1)	7 (36.8)	2 (10.5)	4 (21.1)	2 (10.5)		0.269 ^a^
Other Bleeding	0 (0)	0 (0)	0 (0)	4 (80.0)	1 (20.0)		0.148 ^a^
Intraspinal	0 (0)	0 (0)	0 (0)	0 (0)	0 (0)		
Intraocular	0 (0)	0 (0)	0 (0)	0 (0)	0 (0)		
Intraarticular	0 (0)	0 (0)	0 (0)	0 (0)	0 (0)		
Pericardial	0 (0)	0 (0)	0 (0)	0 (0)	0 (0)		
Intramuscular	0 (0)	0 (0)	0 (0)	0 (0)	0 (0)		

^a^ Fisher’s Exact test; ^b^ Pearson Chi Square.

**Table 5 geriatrics-10-00165-t005:** Logistic regression model with major bleeding as the dependent variable.

		Simple Logistic Regression			Multiple Logistic Regression		
		β	Crude Odd Ratio (95% CI)	*p* Value	β	Adjusted Odd Ratio (95% CI)	*p* Value
Age		0.06	1.06 (1.03, 1.10)	0.001 *	0.07	1.07 (1.03, 1.11)	<0.001 *
Age group	65–74	0	1				
	75–84	0.98	2.67 (1.26, 5.67)	0.010 *			
	≥85	1.23	3.44 (1.53, 7.72)	0.003 *			
Gender	Male	0	1				
	Female	0.01	1.01 (0.61, 1.69)	0.969			
Ethnic	Malay	0	1				
	Chinese	0.25	1.29 (0.75, 2.21)	0.360			
	Indian	−18.49	0	0.998			
	Punjabi	2.71	15.09 (2.03, 112.28)	0.008 *			
	Others	−18.49	0	>0.950			
Concurrent antiplatelet used	No	0	1				
	Yes	1.48	4.41 (0.45, 43.01)	0.202 *			
CKD staging	1–3	0	1				
	4–5	0.33	1.39 (0.53, 3.63)	0.499			
Liver disorder	No	0	1				
	Yes	0.13	1.14 (0.26, 4.95)	0.861			
Hx of major bleeding	No	0	1		0	1	
	Yes	3.72	41.10 (4.21, 401.14)	0.001 *	4.02	55.89 (5.48, 570.45)	0.001 *
Stroke history	No	0	1				
	Yes	−0.61	0.55 (0.23, 1.29)	0.169 *			
Hypertension	No	0	1				
	Yes	0.32	1.38 (0.61, 3.10)	0.436			

* Statistically significant.

**Table 6 geriatrics-10-00165-t006:** Association between HASBLED score and prevalence of major bleeding.

	Major Bleeding, *n* (%)		Statistical Test	*p* Value
	No	Yes		
HASBLED score			23.48 (1) ^a^	<0.001 *
<3	612 (95.5)	29 (4.5)		
≥3	211 (86.1)	34 (13.9)		

* Statistically significant; ^a^ Pearson Chi Square.

**Table 7 geriatrics-10-00165-t007:** Dose appropriateness of each Direct Oral Anticoagulant (DOAC).

	Dose Appropriateness, *n* (%)		*p* Value
	No	Yes	
Apixaban	3 (11.5)	23 (88.5)	0.078 ^a^
Dabigatran	5 (45.5)	6 (54.5)	
Edoxaban	1 (20.0)	4 (80.0)	
Rivaroxaban	10 (35.7)	18 (64.3)	

^a^ Fisher’s exact test.

## Data Availability

The data that support the findings of this study are available on request from the corresponding author. The data are not publicly available due to privacy or ethical restrictions.

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
