# Peer review of "Prevalence of Major Bleeding in Elderly Patients on Oral Anticoagulants for Non-Valvular Atrial Fibrillation: A Single-Center 12-Year Retrospective Review"

_geriatrics, 2025, doi:10.3390/geriatrics10060165_

Round 1
Reviewer 1 Report
Comments and Suggestions for Authors
Kwan FH et al performed a retrospective study assessing bleeding and risk factors for bleeding in patients 65 years or older.
Introduction
– The current introduction provides brief summary about atrial fibrillation but instead the introduction should focus more on prior research in the area of the current studies. Like prior research assessing bleeding risk in this population etc
- Some places says vitamin K antagonists and some say warfarin. Stick to one.
Methodology
- What do authors exactly mean by convenient sampling? How did they recruit patients?
- This paragraph should be re-worded “Dose appropriateness for each DOAC was determined based on guide-95 line-recommended criteria. For apixaban, dose reduction was indicated when two out of 96 three conditions were met: age >80 years, serum creatinine >1.5 mg/dL, or body weight 97 <60 kg. For rivaroxaban, the reduced dose of 15 mg once daily was recommended for 98 patients with CrCl <50 mL/min, while use was not advised if CrCl <15 mL/min. 99 Dabigatran dosing was reduced to 110 mg twice daily in patients over 80 years, to 75 mg 100 twice daily in those with CrCl 15–30 mL/min, and was not recommended for CrCl <15 101 mL/min. For edoxaban, a reduced dose of 30 mg once daily was applied in patients with 102 CrCl 15–50 mL/min or body weight <60 kg”.
- The authors did not give any medications, they rather reviewed charts. For example they should say” The recommended dabigatran dose…”
- Provided a reference for “Japan All Nippon AF in the Elderly (ANAFIE) registry”
- I’m not sure why sample size calculation was performed for a retrospective descriptive study. The study was 80% power for what?
- How did the authors collect outcomes? What if a patient had bleeding at another facility?
Results
- What’s the average follow up time and average time to events? The title says 12 years study but not all patients were followed for 12 years, right?
- What does this mean “middle-old, followed by young-old and old-old groups”?
- For this sentence “Although the 146 absolute number was higher in the DOAC group, the percentage of major bleeding was 147 lower (6.6% vs. 10.5%) due to the larger sample size” the most important is percentage. Please revise to just focus on that
- Few patients had weight data? Were other data complete? What about data on kidney function?
- For this statement “Among the cohort, only 70 patients had recorded body weight (816 patients had 194 missing weight data). Within this subgroup, 88.5% of patients on apixaban were pre-195 scribed an appropriate dose, followed by 80% on edoxaban, 64.3% on rivaroxaban, and 196 54.5% on dabigatran”. But rivaroxaban and dabigatran dose depend more on kidney function and not weight.
Reviewer 2 Report
Comments and Suggestions for Authors
My overall impression of the manuscript: This study addresses an important and timely issue, the occurrence of bleeding adverse events as observed in a real-world setting. The scientific community indeed needs such data. However, if the methodology is not sound and comparable variables are not properly accounted for, the resulting data lose their value and may even be misleading. Therefore, I believe that this study requires substantial methodological improvement before it can be considered for publication.
For example, you say: ”Patients initiated on anti-coagulants for NVAF throughout this period were recruited in this study via convenient sampling method.”. Please explain what ”Convenient sampling method” means, as the manuscript is not clear enough.
Concerning the was did you followed in this study, since you mentioned it is a retrospective cohort, with patient follow-up performed as part of their routine care, you cannot measure the prevalence, you measure the incidence of bleeding events. Period prevalence is measured in a cross-sectional study. It is a confusion here. Please clarify that. The incidence does not take into account the duration of exposure to the anticoagulant (which may lead to bias of interpretation). In other words, a patient who started anticoagulant therapy (most likely warfarin) in 2012 and experienced a bleeding event in 2021 (after 9 years of treatment) is considered equivalent in your study to a patient who started treatment in 2021 (more likely a DOAC) and had a bleeding event in 2022 (after only 1 year of treatment). The chi-squared test is useful for comparing the proportion of patients who suffered major bleeding, but the picture is not complete. I suggest evaluating the duration of treatment and express the risk by number-of-events/person-time for example, because longer exposed patients have a higher risk of events, so simple comparation induces bias. You may extract the duration of treatment as a variable and analyze using incidence rate ratio or hazard ratio (if you want cu adjust on age, sex, comorbidities, etc) or Kaplan_meier curves with log-rank test (to compare warfarine and DOAC).
Round 2
Reviewer 1 Report
Comments and Suggestions for Authors
Thank you for revising the manuscript.
When revising, it's important to note the page and lines or the new change.
Some comments not fully address:
- What's the average follow up time?
- How can you determine dose appropriateness without weight data?
- How many hospitals are in the region? How common/uncommon is it for patients to seek care at other facilities?
Reviewer 2 Report
Comments and Suggestions for Authors
The answers to my first review and the changes are satisfactory.
